

# Reliability of the use of foot pressure pain threshold in adults: a test-retest analysis

Lidia Mayorga-Vega[1,2], Ana Maria Jimenez-Cebrian[1,2],
Francisco Javier Barón-López[2,3], Alonso Montiel-Luque[1,4],
Juan Carlos Benavente-Marín[1,2] and Julia Warnberg[1,2]

[1] Department of Nursing and Podiatry, University of Málaga, Málaga, Spain
[2] EpiPHAAN Research Group, Universidad de Málaga—Instituto de Investigación Biomédica de Málaga (IBIMA-BIONAND), Málaga, Spain
[3] Department of Public Health, University of Málaga, Málaga, Spain
[4] San Miguel Health Center, Costa del Sol Primary Healthcare District, Málaga, Spain

Corresponding author
Ana Maria Jimenez-Cebrian,
amjimenezc@uma.es

## ABSTRACT

**Background:** Foot pain is very common, affecting 24% of the adult population over 45 years old. Mechanical stimulation devices, such as pressure algometers, are used for the precise and objective quantification of the pain threshold (minimum tolerable pain). The digital pressure algometer is a diagnostic tool for evaluating pain thresholds and pain tolerance related to musculoskeletal conditions in patients. Pressure pain thresholds (PPT) measured with a digital algometer have shown good to excellent intra-rater agreement and reliability for most body locations. However, no study has focused on evaluating specific locations on the foot. This study aimed to determine the intra- and inter-session agreement and reliability of pressure pain threshold measurements on the foot in a population of adults aged 55 to 75 years.
**Method:** Using a JTECH Commander digital algometer, an experienced podiatrist measured the pressure pain thresholds at four locations on the sole of the dominant and non-dominant foot (head of the first metatarsal, head of the fifth metatarsal, pulp of the first toe, and center of the heel) in 15 male and 16 female volunteers aged 55 to 75. In each session, three pressure pain thresholds measurements were taken at each location. Measurements were repeated over three different sessions; the first and second sessions were 30 min apart, and the third session was 1 week later. The standard error of measurement (SEM) and coefficient of variation (CV) were calculated to reflect within- and between-session agreement. Reliability was assessed using intraclass correlation coefficients (ICC).
**Results:** Within-session agreement, expressed as SEM, ranged from 3Newtons (N) to 8N depending on the test location. Between session agreement, expressed as SEM, ranged from 5N to 10N, and the CV ranged from 7.9% to 13.7% per location. The ICC values ranged from 0.76 to 0.91.
**Conclusions:** A digital algometer can be used to measure pressure pain thresholds on the foot. Specifically, this study used this technique to evaluate PPT at four points on the plantar surface (head of the first and fifth metatarsals, pulp of the first toe, and center of the heel), and the results showed good to excellent reliability for within- and between-session measurements.

# INTRODUCTION

According to the International Association for the Study of Pain, pain is defined as "an unpleasant sensory and emotional experience associated with actual or potential tissue damage" (*Raja et al., 2020*). Pain can be classified based on its duration as either acute or chronic. Acute pain has limited duration with minimal psychological influence, while chronic pain persists indefinitely and is accompanied by a psychological component (*Puebla Díaz, 2005*).

Foot pain is very common, affecting 24% of adults over the age of 45 (*Menz et al., 2013*). Studies have shown that people suffering from foot pain experience a reduced quality of life, as it may hinder daily activities, cause balance and walking problems, and increase the risk of falls, especially in older individuals (*Laslett et al., 2018*; *Cotchett et al., 2022*).

Pain is difficult to measure and can be evaluated subjectively (as reported by the individual) or objectively (quantified after being triggered). The Visual Analog Scale (VAS) is commonly used for subjective pain measurement, assessing the intensity of pain as perceived by the individual (*DeLoach et al., 1998*). Precise and objective quantification of the pain threshold (the minimum tolerated pain) is crucial for many aspects of pain management, from clinical research to evaluating treatment efficacy. Mechanical stimulation devices, such as pressure algometers, are used for this purpose (*Fischer, 1987*). An algometer displays the pressure pain threshold (PPT), which is the minimum pressure applied to the evaluated area that triggers pain (*Fischer, 1987*; *Kinser, Sands & Stone, 2009*; *Saban & Masharawi, 2016*). It is important to note that the reliability of pressure pain threshold data depends on the technique used by the observer and the subject's ability to provide clear verbal feedback on their perception of pressure pain thresholds levels (*Chesterton et al., 2007*). Within the framework of Quantitative Sensory Testing (QST), a standard protocol is used to measure pressure pain thresholds. This involves using a digital algometer to collect multiple PPT measurements at target points to obtain an average value, as well as factoring in variables such as patient positioning and applied pressure force to enhance reproducibility. The procedure is standardized by providing precise instructions and training for data collection (*Rolke et al., 2006*). This standardization helps minimize variability and ensures consistent results across different evaluators and sessions.

Intra-rater reproducibility refers to the degree of agreement between pressure pain threshold measurements taken by the same evaluator at different times. This reproducibility can be divided into two test-retest parameters: agreement and reliability. Agreement (or measurement error) refers to the magnitude of difference between pressure pain threshold measurements taken during the same session and those taken in different sessions. Agreement is expressed as an absolute value such as the standard error of measurement (SEM) or the coefficient of variation (CV). Reliability refers to the evaluator's ability to distinguish between individuals based on their pressure pain

thresholds and is expressed as a relative value such as the intraclass correlation coefficient (ICC) (*Chesterton et al., 2007*; *Fischer, 1987*; *Kinser, Sands & Stone, 2009*; *Nussbaum & Downes, 1998*; *Walton et al., 2011*).

Previous studies have demonstrated the reliability and validity of algometer measurements in different parts of the body (*Chesterton et al., 2007*; *Fischer, 1987*; *Kinser, Sands & Stone, 2009*; *Walton et al., 2011*), but there is a lack of data from individuals with healthy feet without previous pathologies. The digital pressure algometer is a diagnostic tool used to assess pain thresholds in patients, especially for musculoskeletal conditions like fibromyalgia, myofascial pain, and joint discomfort. The digital algometer also has several advantages, such as being easy to use and providing accurate and reliable measurements (*Jones, Kilgour & Comtois, 2007*; *Nussbaum & Downes, 1998*). The digital algometer is frequently used to measure pain thresholds in the head (*Castien et al., 2021*), tongue (*Díaz-Saez et al., 2023*), and neck (*Knapstad et al., 2018*). Other studies have used the digital algometer to evaluate pain thresholds in patients with rheumatoid arthritis (*Huskisson & Hart, 1972*), specifically in the lower limbs at the knee level (*Stausholm et al., 2023*), where the algometer showed good intra- and inter-rater reliability for measuring knee pain thresholds in arthritis patients.

In previous studies on foot pain, algometers have been used to assess pain thresholds before and after treatment of chronic heel pain due to calcaneal spurs (*Yürük, Aykurt Karlıbel & Kasapoğlu Aksoy, 2022*). Pressure pain thresholds may be an indicator of the effects of foot posture throughout the workday. Greater changes in pressure pain thresholds and increased foot discomfort were found in workers who spent more hours standing (*Messing & Kilbom, 2001*). Pain sensitivity varies according to the area of the foot—higher in the arch and dorsum of the foot and lower in the heel—which has a significant impact on the design of footwear (*Xiong, Goonetilleke & Jiang, 2011*). Algometers have also been used to measure treatment outcomes for plantar fasciitis (*Yelverton, Rama & Zipfel, 2019*), providing results before and after treatments (*Boob, Phansopkar & Somaiya, 2023*). *Madeleine et al. (2014)* used pressure pain threshold measurements to demonstrate the effect of cushioning footwear inserts. Additionally, pain measurement can be useful for tracking its progression over time or as a predictive tool for future outcomes (*Walton et al., 2011*).

However, no study has evaluated pressure pain thresholds of the feet in asymptomatic individuals. This study aimed to determine the intra- and inter-session agreement and reliability of foot measurements using pressure pain threshold in adults aged 55 to 75 years.

## MATERIALS AND METHODS

### Study design and participants

This study aimed to examine the reliability of a diagnostic test in an adult population aged 55 to 75 years. This age range was chosen because the primary purpose of this study was to test the reliability of the digital algometer in measuring foot pain in a population similar to

the cohort that will participate in a larger project (the PREDIMEDPlus study, which includes participants from Málaga within this same age range). Participants were selected through convenience sampling. Measurements were taken from June 10 to June 17, 2024. Participants were excluded if they had any of the following conditions: amputation of one or both lower limbs, ulcers or open wounds on the soles of the feet, calcaneal spurs, plantar fasciitis (acute foot pain), or were undergoing systemic corticosteroid treatment. This study was conducted and reported in accordance with the Guidelines for Reporting Reliability and Agreement Studies (GRAAS) (*Kottner et al., 2011*).

Patients regularly attending a podiatry clinic were invited to participate in the study through a phone call, during which they were asked to attend an informational meeting. At this meeting, the purpose of the study was explained in detail, and participants were given a written informed consent form to sign. All participants provided written informed consent before participating in the study. The measurements were conducted at the Lidia Mayorga Vega Podiatry Clinic located in Puerto de la Torre (Málaga, Spain).

### Sample size

A minimum of 25 participants was required to ensure reliable calculations based on an alpha level of 0.05, power (P) of 0.8, and an expected ICC greater than 0.85 (with a lower bound of the 95% confidence interval at 0.65). Given that the study involved two appointments spaced 1 week apart and that there was a potential risk of participant drop-out between sessions, 32 volunteers were initially recruited to secure the desired sample size. In total, 31 volunteers attended both sessions.

### Procedure

A total of 31 patients were included in the study. Body Mass Index (BMI) was calculated by dividing weight in kilograms by the square of height in meters ($kg/m^2$).

The pressure algometer was used to objectively assess foot pain threshold, defined as the minimum tolerable pressure before pain is perceived (*Fischer, 1987*; *Trueba-Perdomo, Gasparini & Cuautle, 2021*). The specific algometer used for these measurements was the Algometer Commander from JTECH Medical, manufactured in 2004. This instrument featured a 1 $cm^2$ probe for measurement, using Newtons (N) as the unit of measurement (*Saban & Masharawi, 2016*).

During the procedure, the patient was positioned comfortably in a supine position on an examination table. Measurements were taken on both feet, the right (R) and the left (L), at four specific points on the plantar aspect: the first metatarsal head (1st MH), the fifth metatarsal head (5th MH), the first toe pulp (1st TP), and the heel. These correspond to the areas of the foot that typically experience the greatest contact with the ground throughout the different phases of gait. For each point, the probe was placed perpendicularly to the foot, and pressure was applied gently to the target area (*Castien et al., 2021*). The process was halted as soon as the patient reported any discomfort. Measurements at each point were taken three times, with the same procedure repeated during the initial trial, after 30 min, and again after 7 days. The podiatrist who performed the measurements had previous clinical experience using the algometer in routine practice for the assessment of

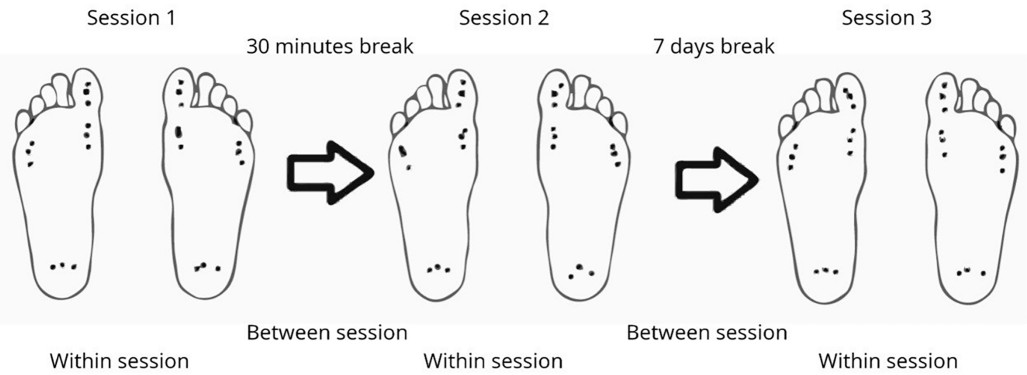

**Figure 1 Visual representation of the methodology.** The images illustrate which measurements are used for calculations within and between sessions. In each session, the four points (1st metatarsal head, 5th metatarsal head, 1st toe pulp, and heel) are measured on both the left and right feet, with three measurements taken at each point. The time interval between sessions 1 and 2 is 30 min, and between sessions 2 and 3, it is 7 days.

pressure pain thresholds. In addition, prior to the study, the assessor underwent a standardized process of calibration and familiarization with the device to ensure consistency of measurements. Figure 1 presents a visual representation of the methodology used in this study, illustrating the pressure areas measured and the parameters calculated both within and between sessions.

## Ethical approval

This study adhered to the principles outlined in the Declaration of Helsinki (*World Medical Association, 2025*). The Ethics Committee for Research at the University of Málaga approved the study protocol under registration number 199-2023-H (CEUMA).

## Descriptive statistics

Descriptive analyses were used to summarize participant characteristics including age, sex and BMI. Continuous variables were reported as means and standard deviations (SD), and categorical variables as frequencies and percentages.

The assumption of normality was assessed visually using quantile-quantile (Q-Q) plots and statistically with the Shapiro-Wilk test. Normally distributed data were presented as mean (SD), while non-normally distributed variables were described using medians and interquartile ranges.

Comparisons between sexes were made using independent t-test or Mann-Whitney U test, as appropriate. The Chi-squared test was used for categorical variables. Additionally, the mean PPT at each location and sesion was derived from the average of the three session per location.

## Reproducibility parameters

The standard error of measurement (SEM) and coefficient of variation (CV) intra-rater parameters were calculated for both within-session and between-session agreement. The intraclass correlation coefficient (ICC) was calculated to assess intra-rater reliability. As the

**Table 1 Participant characteristics.**

| Characteristic | Overall, $N = 31$[1] | Women, $N = 16$[1] | Men, $N = 15$[1] |
|---|---|---|---|
| Age (year) | 65.8 (5.2) | 64.1 (5.0) | 67.6 (4.9) |
| Height (cm) | 164.1 (5.9) | 160.7 (5.8) | 167.7 (3.5) |
| Weight (kg) | 78.8 (11.8) | 76.3 (9.9) | 81.5 (13.3) |
| BMI (kg/m²) | 29.3 (3.9) | 29.6 (3.8) | 28.9 (4.2) |
| BMI (three categories) | | | |
| Normal weight | 5/31 (16%) | 2/16 (13%) | 3/15 (20%) |
| Overweight | 14/31 (45%) | 8/16 (50%) | 6/15 (40%) |
| Obesity | 12/31 (39%) | 6/16 (38%) | 6/15 (40%) |

Notes:
Abrevations: BMI, Body Mass Index; Normal weight (BMI = 22–25), Overweight (BMI = 25–30), Obesity (BMI > 30).
[1] Mean (SD); n/N (%).
[2] Two Sample t-test.

**Table 2 Pressure pain thresholds at baseline.**

| Characteristic | Overall, $N = 31$[1] | Women, $N = 16$[1] | Men, $N = 15$[1] |
|---|---|---|---|
| 1st MH, R (N) | 71.7 (21.2) | 68.7 (21.0) | 74.9 (21.8) |
| 5th MH, R (N) | 72.4 (23.3) | 69.0 (23.9) | 76.0 (22.9) |
| 1st TP, R (N) | 45.1 (19.0) | 39.8 (15.7) | 50.8 (21.0) |
| Heel, R (N) | 91.9 (20.9) | 90.6 (21.7) | 93.3 (20.6) |
| 1st MH, L (N) | 72.6 (20.1) | 68.5 (19.6) | 77.0 (20.3) |
| 5th MH, L (N) | 73.6 (20.9) | 69.7 (19.9) | 77.7 (21.8) |
| 1st TP, L (N) | 48.3 (19.6) | 43.5 (17.7) | 53.4 (20.8) |
| Heel, R (N) | 91.5 (21.5) | 91.2 (22.1) | 91.8 (21.6) |

Notes:
L, Left foot; R, Right foot; 1st MH, first metatarsal head; 5th MH, fifth metatarsal head; 1st TP, first toe pulp; H, heel; N, Newton; CV, coefficient of variation.
[1] Mean (SD).
[2] Two Sample t-test.

study involved a single fixed evaluator conducting the same measurements multiple times, the ICC (two-way mixed effects model: 3, 1) was selected. This model is appropriate for assessing the consistency of repeated measurements by a fixed rater. Calculations were performed following the guidelines provided by *Koo & Li (2016)* and *Mondal et al. (2024)*. The minimal detectable change at 90% confidence (MDC90) was calculated based on the SEM. All reproducibility statistics were calculated using R package psych (*Revelle, 2024*).

## RESULTS

A total of 31 participants were included in this study. Of these participants, 16 were women and 15 were men, with a mean age of 64.1 and 67.6 years, respectively. The average BMI was 29.6 kg/m² for women and 28.9 kg/m² for men. Regarding BMI categories, 12.5% of women and 20% of men were classified as normal weight, 50% of women and 40% of men as overweight, and 37.5% of women and 40% of men as obese (Table 1). Pressure pain threshold values from session 1 are shown in Table 2. All intra-session and inter-session agreement and reliability values are presented in Tables 3 and 4 and are illustrated in Fig. 2.

| Table 3 Description of data within sessions. | | | | | |
|---|---|---|---|---|---|
| Session | Location | PPT (mean N) | SEM (N, 95% CI) | CV (95% CI) | ICC (95% CI) |
| 1 | 1stMH, R | 71.72 | 8 [6–9] | 10.6% [8.9–12.8%] | 0.88 [0.80–0.94] |
| 2 | 1stMH, R | 68.86 | 6 [5–8] | 9.2% [7.7–11.1%] | 0.92 [0.87–0.96] |
| 3 | 1stMH, R | 69.04 | 5 [4–6] | 7.6% [6.4–9.2%] | 0.92 [0.86–0.96] |
| 1 | 1stMH, L | 72.61 | 8 [6–9] | 10.5% [8.8–12.7%] | 0.87 [0.78–0.93] |
| 2 | 1stMH, L | 68.68 | 5 [4–6] | 6.9% [5.8–8.4%] | 0.95 [0.92–0.97] |
| 3 | 1stMH, L | 70.22 | 5 [4–6] | 7.4% [6.2–8.9%] | 0.92 [0.87–0.96] |
| 1 | 5thMH, R | 72.42 | 7 [6–8] | 9.6% [8.1–11.6%] | 0.92 [0.85–0.96] |
| 2 | 5thMH, R | 70.06 | 6 [5–8] | 8.8% [7.4–10.7%] | 0.91 [0.85–0.95] |
| 3 | 5thMH, R | 75.00 | 5 [4–6] | 6.6% [5.5–7.9%] | 0.92 [0.87–0.96] |
| 1 | 5thMH, L | 73.55 | 7 [6–9] | 9.9% [8.3–12.0%] | 0.89 [0.81–0.94] |
| 2 | 5thMH, L | 70.85 | 8 [7–10] | 11.4% [9.6–13.8%] | 0.86 [0.76–0.92] |
| 3 | 5thMH, L | 76.10 | 5 [4–6] | 6.7% [5.6–8.1%] | 0.92 [0.85–0.96] |
| 1 | 1stTP, R | 45.11 | 5 [4–6] | 10.2% [8.6–12.3%] | 0.94 [0.90–0.97] |
| 2 | 1stTP, R | 48.85 | 4 [4–5] | 8.7% [7.3–10.5%] | 0.95 [0.90–0.97] |
| 3 | 1stTP, R | 51.21 | 3 [3–4] | 6.4% [5.3–7.7%] | 0.96 [0.93–0.98] |
| 1 | 1stTP, L | 48.33 | 4 [4–5] | 9.3% [7.8–11.3%] | 0.95 [0.91–0.97] |
| 2 | 1stTP, L | 50.52 | 4 [4–5] | 8.7% [7.3–10.6%] | 0.94 [0.90–0.97] |
| 3 | 1stTP, L | 51.16 | 3 [3–4] | 6.1% [5.1–7.3%] | 0.97 [0.94–0.98] |
| 1 | Heel, R | 91.92 | 6 [5–7] | 6.2% [5.2–7.5%] | 0.92 [0.88–0.96] |
| 2 | Heel, R | 89.75 | 4 [3–4] | 3.9% [3.3–4.7%] | 0.97 [0.95–0.99] |
| 3 | Heel, R | 91.91 | 5 [4–6] | 5.0% [4.2–6.0%] | 0.93 [0.88–0.97] |
| 1 | Heel, L | 91.45 | 6 [5–7] | 6.4% [5.4–7.7%] | 0.93 [0.88–0.96] |
| 2 | Heel, L | 89.30 | 6 [5–7] | 6.5% [5.5–7.9%] | 0.92 [0.87–0.96] |
| 3 | Heel, L | 90.72 | 4 [4–5] | 4.7% [4.0–5.7%] | 0.95 [0.92–0.98] |

**Note:**
L, Left foot; R, Right foot; 1st MH, first metatarsal head; 5th MH, fifth metatarsal head; 1st TP, first toe pulp; H, heel; N, Newton; SEM, Standard Error of Measurement; CV, coefficient of variation; ICC, Reliability Interval.

| Table 4 Description of data between sessions. | | | | | |
|---|---|---|---|---|---|
| Location | PPT (mean N) | SEM (N, 95% CI) | MDC 90% (N, 95% CI) | CV (95% CI) | ICC (95% CI) |
| 1stMH, R | 69.87 | 7 [6–9] | 17 [15–21] | 10.7% [9.0–13.0%] | 0.87 [0.78–0.93] |
| 1stMH, L | 70.51 | 8 [7–9] | 18 [15–22] | 11.0% [9.2–13.3%] | 0.85 [0.75–0.92] |
| 5thMH, R | 72.49 | 10 [8–12] | 23 [20–28] | 13.7% [11.5–16.6%] | 0.77 [0.62–0.87] |
| 5thMH, L | 73.50 | 8 [7–10] | 20 [17–24] | 11.5% [9.6–13.9%] | 0.81 [0.69–0.90] |
| 1stTP, R | 48.39 | 5 [5–7] | 12 [11–15] | 11.1% [9.3–13.4%] | 0.90 [0.84–0.95] |
| 1stTP, L | 50.00 | 5 [5–6] | 12 [10–15] | 10.6% [8.9–12.8%] | 0.91 [0.85–0.95] |
| Heel, R | 91.19 | 7 [6–9] | 17 [14–20] | 7.9% [6.6–9.5%] | 0.87 [0.78–0.93] |
| Heel, L | 90.49 | 8 [7–10] | 19 [16–23] | 9.1% [7.7–11.0%] | 0.84 [0.73–0.91] |

**Note:**
L, Left foot; R, Right foot; 1st MH, first metatarsal head; 5th MH, fifth metatarsal head; 1st TP, first toe pulp; H, heel; N, newton; SEM, Standard Error of Measurement; CV, coefficient of variation; MDC, Minimal detectable change; ICC, Reliability Interval.
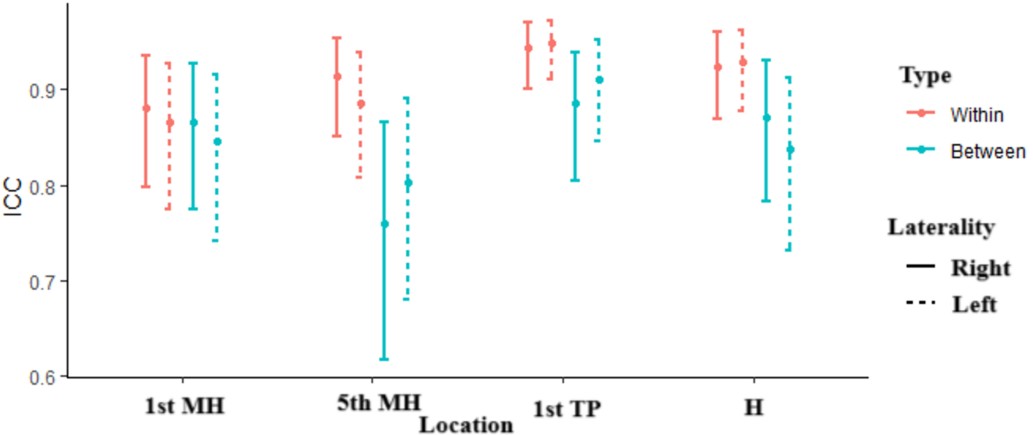

**Figure 2  ICC location.** The figure shows the ICCs for four different locations: the first metatarsal head (1st MH), fifth metatarsal head (5th MH), 1st toe pulp (1st TP), and heel (H), across both within-session (red) and between-session (blue) comparisons. The solid lines represent measurements taken on the right foot (R), while the dashed lines indicate those on the left foot (L). The error bars depict the confidence intervals for each ICC value, illustrating the reliability of measurements within and between sessions for each location.                         

## Pressure pain thresholds and within-session agreement and reliability

Intra-session agreement is presented in Table 3. The SEM within sessions ranged from 3N to 8N, depending on the test location and session. The CV within sessions ranged from 3.9% to 13.7%. CV values varied between 6.2% and 10.6% in session 1, 3.9% and 11.4% in session 2, and 4.7% and 7.6% in session 3.

The ICC values for within-session reliability were good to excellent, ranging from 0.86 to 0.97, indicating a high level of consistency in repeated measurements taken by the same evaluator.

## Between-session agreement and reliability

Inter-session agreement is presented in Table 4. Between-session agreement was also assessed using SEM and CV. SEM values between sessions ranged from 5N to 10N, and CV values ranged from 7.9% to 13.7%, depending on the location. The ICC values for between-session reliability were slightly lower compared to within-session reliability, ranging from 0.76 to 0.91, but these values still indicated acceptable reliability. SEM values were 7N for the 1st MH (L) and 8N for the 1st MH (R), 10N for the 5th MH (L) and 8N for the 5th MH (R), 5N for both 1st TPs (L/R), and 7N for the heel (L) and 8N for the heel (R). The CV values were 10.7% for the 1st MH (L) and 11% for the 1st MH (R), 13.7% for the 5th MH (L) and 11.5% for the 5th MH (R), 11.1% for the 1st TP (L) and 10.6% for the 1st TP (R), and 7.9% for the heel (L) and 9.1% for the heel (R).

## Individual variation

Individual variation in pressure pain threshold measurements was observed across the different foot locations. The CV values indicated variability in the reproducibility of

measurements, with some locations *e.g.*, 5th MH) showing higher variation (up to 13.7%) compared to others (*e.g.*, heel, with 7.9%). The minimal detectable change (MDC) at a 90% confidence level ranged from 12N to 23N across the different locations, reflecting the magnitude of individual variation in pain threshold measurements.

## DISCUSSION

This study was designed to examine the intra-rater reliability of the Commander JTECH digital algometer (2004) for pain pressure threshold measurements taken from the foot. The results showed good to excellent reliability (ICC: 0.76 to 0.91), supporting the hypothesis that the digital algometer is a reliable tool for measuring pressure pain thresholds on the foot, similar to its use on other parts of the body as shown in previous studies. Furthermore, the device provides consistently reliable results, making it a useful tool for both clinical practice and research.

Several previous studies have examined the reliability of pressure pain threshold (PPT) measurements on the foot, although with differing methodologies and objectives. *Xiong, Goonetilleke & Jiang (2011)* reported an ICC of 0.8 when evaluating pressure thresholds in the right foot of 20 participants, with results influenced by stimulus characteristics such as probe size and indentation speed. However, unlike the current study, their focus was on mechanical variables relevant to product design rather than clinical assessment using standard tools. Similarly, *Wu, Liu & Qu (2024)* used algometry to map foot regions in 20 adults and reported regional differences in pain sensitivity, with lower thresholds in the dorsum and lateral areas and higher thresholds in the plantar region, particularly the heel. While informative, these studies did not assess intra-rater reliability under clinical conditions. In contrast, *Saban & Masharawi (2016)* reported ICCs ranging from 0.75 to 0.92 when testing five heel regions, which aligns with the reliability values obtained in our study. Furthermore, *Jerez-Mayorga et al. (2020)* and *Reezigt et al. (2023)* confirmed the algometer's reliability and usability, including among inexperienced raters, though they focused on body areas beyond the foot. Therefore, our study adds to the literature by being, to our knowledge, the first to evaluate the intra-rater reliability of a digital pressure algometer applied systematically across specific foot regions in an older adult population, offering clinically relevant insights and supporting the instrument's applicability in routine assessments.

The findings of the current study support the use of the digital algometer as a fundamental tool for measuring pressure pain thresholds. Its demonstrated accuracy, intra-rater reliability, and ease of use make it a valuable instrument for clinical assessment of pain sensitivity. These characteristics enhance its utility in monitoring patients' responses to interventions over time and in ensuring consistency in pain evaluations, thereby contributing to more precise clinical decision-making.

### Limitations

This study has several limitations that should be considered when interpreting its results. First, the sample consisted exclusively of older adults, representing a heterogeneous group, which limits the generalizability of the findings to other populations such as children or

young adults. In addition, the evaluator was not completely blinded to all the measurements made during the sessions, which could have generated a bias in their observations or interpretations because of their prior knowledge of certain aspects of the study. Also, as the measurements were made by a single evaluator, it was not possible to assess interobserver reliability, a relevant aspect that should be addressed in future research. These limitations underscore the importance of conducting additional studies with more heterogeneous subject samples and with the participation of multiple raters to ensure the consistency of the measurements in a broader context.

## CONCLUSIONS

The digital algometer can be used to measure pressure pain thresholds in the healthy foot. Specifically, this study evaluated pain in four points on the plantar surface of the foot (the head of the 1st and 5th metatarsals, the pulp of the 1st toe, and the center of the heel), resulting in good to excellent reliability.

### Funding
The authors received no funding for this work.

### Competing Interests
Julia Wärnberg and Ana Maria Jimenez-Cebrian are Academic Editors for PeerJ.

### Author Contributions
- Lidia Mayorga-Vega conceived and designed the experiments, performed the experiments, authored or reviewed drafts of the article, and approved the final draft.
- Ana Maria Jimenez-Cebrian conceived and designed the experiments, performed the experiments, authored or reviewed drafts of the article, and approved the final draft.
- Francisco Javier Barón-López conceived and designed the experiments, analyzed the data, prepared figures and/or tables, and approved the final draft.
- Alonso Montiel-Luque conceived and designed the experiments, analyzed the data, prepared figures and/or tables, and approved the final draft.
- Juan Carlos Benavente-Marín conceived and designed the experiments, prepared figures and/or tables, and approved the final draft.
- Julia Warnberg conceived and designed the experiments, analyzed the data, prepared figures and/or tables, authored or reviewed drafts of the article, and approved the final draft.

### Human Ethics
The following information was supplied relating to ethical approvals (*i.e.*, approving body and any reference numbers):

Ethics Committee of the University of Malaga approved the study (registration number CEUMA 199-2023-H).

## Ethics

The following information was supplied relating to ethical approvals (*i.e.*, approving body and any reference numbers):

The Ethics Committee for Research at the University of Málaga approved the study protocol under registration number CEUMA 199-2023-H.

## Data Availability

Raw data is available in the Supplemental Files.

## Supplemental Information

Supplemental information for this article can be found online at http://dx.doi.org/10.7717/peerj.19875#supplemental-information.

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
