# Peer review of "Reliability of the use of foot pressure pain threshold in adults: a test-retest analysis"

_PeerJ, doi:10.7717/peerj.19875_

## Round 0.1 · original submission · Major Revisions

Both reviewers have several important concerns which you must address. In particular please be sure to deal with Reviewer 1's concerns regarding the selection of the group aged 55-75.

Reviewer 1 ·

Basic reporting

No comment

Experimental design

The manuscript titled "Reliability of the use of Foot Pressure Pain Threshold in adults: A test-retest analysis" addresses an important topic by evaluating the intra-rater reliability of a digital pressure algometer for assessing pressure pain thresholds (PPTs) at specific foot locations. The study provides valuable insights into the reproducibility of PPT measurements, showing good to excellent reliability (ICC: 0.76–0.91) across test locations.

Validity of the findings

Major issues
*The authors studied an elderly population aged 55-75. It is unclear what the rationale for selecting this population was. These kinds of psychophysical studies have merit when establishing normal ranges in either healthy participants or participants with pathologies. The demographics provided, along with the narrow inclusion criteria, raise the suspicion that this is not necessarily a healthy population, with BMI averages showing most were overweight. At these ages and weights, people tend to have diabetes and take many medications like blood thinners, beta blockers etc. Also, at these ages, many patients also can be cancer survivors after chemotherapy. With all these things not recorded and documented, it is difficult to ascertain the true nature of the population. They could be considered healthy, but they could also be considered heterogeneous unhealthy. These points make conclusion-making impossible.

Additional comments

Minor issues
*The authors should note that PPT is actually a misnomer and that the algometers test is pain-inducing stimulus intensity. There isn’t necessarily a connection between clinical pain and the result of the algometer test. If someone has plantar fasciitis and you do this test it does not asses the patient’s clinical pain but merely the stimulus intensity that provokes a report of pain in this condition. This theme is confused across the manuscript in the intro and discussion. Please see this ref:
Cohen, Milton, Asaf Weisman, and John Quintner. "Pain is not a “thing”: how that error affects language and logic in pain medicine." The Journal of Pain 23.8 (2022): 1283-1293.
*The last paragraph of the introduction should clearly state the aim of assessing test-retest reliability in the population of.....?healthy older adults? (which might not be the case)
*There's ambiguity in how the overall mean PPT values were calculated. The manuscript should clarify:
• Whether they used raw measurements (all 9 measurements)
• Or averaged session means (average of 3 session means)
• Or averaged within-session means first
This would affect both the mean values and their associated confidence intervals and should be explicitly stated in the statistical methods section.
*The discussion is too thin and lacks clinical meaning and implications.

Reviewer 2 ·

Basic reporting

Please see below for detailed comments.

Experimental design

Please see below for detailed comments.

Validity of the findings

Please see below for detailed comments.

Additional comments

1. Title:
The title clearly reflects the outcomes measured in the study.

2. Abstract:
The abstract reflects the content of the article.

3. Introduction:
Overall, the introduction includes the necessary information for the reader. It states the problem being investigated and provides a justifiable rationale for the study. However, the authors need to carefully refine the structural organisation of the paragraphs and their content. The sequencing of the sentences/concepts presented are disorganised making this section difficult to read (e.g. L73 stats that the pressure algometer is used in different populations, and this is then repeated in L103 to 105. Further, the final sentence in paragraph 3 appears out of sync).
There are also some specific areas for the authors to consider revising/reviewing:
L61: please insert reference for the sentence regarding foot pain prevalence. L64: the reference used demonstrates the association between foot pain and mobility (only). The authors should include additional references for the associations with balance, walking problems, falls.
L100-113: This paragraph would benefit if there was increased focus on PPTs for feet given that is what the current study is about. For example, in L109-112, you refer to previous studies that have used PPTs for foot pain. Please consider expanding on these reporting if reproducibility was investigated and the main findings.

4. Methodology:
Did you report the study according to reporting guidelines for reproducibility studies (GRRAS checklist: https://pubmed.ncbi.nlm.nih.gov/21130355/). Please check against this checklist and confirm this stating you reported the study according to the guidelines. Please submit the checklist as a supplementary file.
L128: provide dates of recruitment of participants. Please state detailed inclusion criteria for participants. Were participants with foot or lower limb pain excluded (or was it just those with spurs/fasciitis)
L149-150: the authors use the study by Kinser 2009 to define how they defined the perception of pain by participants. However, the study by Kinser 2009 did not use humans – was there a specific methodological approach that was used that you based your PPT method on? I think it is critical to present the specific wording used to instruct participants when to advise they perceived pain. Was the probe 1cm or 1cm squared?
L155-161: I would be interested to understand why the assessors performed measures on both feet. Three measures were taken for each point of application – was the assessor aware of the values for each measure or were they blinded? Can you please elaborate on why you chose the specific points for application of the algometer. Was the application to the plantar aspects of the 1st and 5th MH? Was the toe allowed to move when application was to the toe-pulp? Please consider providing a figure (photo[s]) to illustrate the application of PPTs.
L159: I am unsure what the purpose of the Duenas 2021 reference is. Please revise the text to make the use of this reference more obvious. Did you attempt to standardize the rate of application of force?
Please can the authors describe who the assessors were – how many, their experience, and their training in using the PPT methods described in the study. This is important for understanding the generalisability of the findings.
L183: The authors chose an ICC 1,1 model for analysis. ‘In this model, each subject is rated by a different set of raters who were randomly chosen from a larger population of possible raters. Practically, this model is rarely used in clinical reliability analysis because majority of the reliability studies typically involve the same set of raters to measure all subjects.’ (Koo and Li, J Chiropr Med. 2016 Mar 31;15(2):155–163). Therefore, please can you carefully check your chosen ICC model to be certain the correct analysis has been applied. Please state the software used to calculate your reproducibility statistics.


5. Results:
L197: please consider deleting this sentence.
L213-217: the sentences ‘SEM values…’ and ‘The CV values…’ are not needed as Table 4 shows the data.
Table 1: please consider removal of p-values comparing sexes – this is not a study aim (and remove from description in methods).
Table 2: I am unsure why CV values have been presented here? please remove p-values comparing sexes as this is not a study aim. What is ‘baseline’ and how is this different from ‘session 1’ reported in Table 3?
Table 4: I am unsure why ‘PPT mean N’ values have been reported here for b/w sessions (when they have been reported for individual sessions in Table 3).


6. Discussion/Conclusion:
L232 and 234: the study showed that the PPT tool was reproducible. It did not demonstrate it was ‘valid’. Please revise these sentences.
The discussion is overly brief for a scientific manuscript and needs language edits. Please consider a more detailed paragraph two where you compare and contrast the findings to previous studies that have evaluated PPTs in the foot. (I don’t think you need to include studies of the upper limb).
Please include a more detailed paragraph three where you discuss the clinical implications of the findings – what specific applications are there for measuring PPTs?
The limitations paragraph is overly brief. I agree that the findings may not be generalisable to populations with foot pain as you recruited asymptomatic participants. Further, I agree that the findings may not be generalisable as they are based on one assessor. Are there other limitations? For example, was the assessor blind to all measures taken within a session? Would you obtain high reproducibility if you used different foot landmarks? What future research is needed?

---

## Round 0.2 · Minor Revisions

The reviewer is generally positive about the manuscript but has some minor requests for further changes.

**Language Note:** The review process has identified that the English language must be improved. PeerJ can provide language editing services - please contact us at [email protected] for pricing (be sure to provide your manuscript number and title). Alternatively, you should make your own arrangements to improve the language quality and provide details in your response letter. – PeerJ Staff

Reviewer 2 ·

Basic reporting

Please refer to section 4 below.

Experimental design

Please refer to section 4 below.

Validity of the findings

Please refer to section 4 below.

Additional comments

Thank you for responding to my queries. My queries regarding the revised manuscript are below.

Abstract:
I cannot see any work in the presented study to evaluate ‘validity’. Please remove referring to it from the abstract.

Introduction
Please consider moving this sentence ‘The sensitivity to pain varies according to the area of the foot, being higher in the arch and dorsum of the foot and lower in the heel, having a significant impact on the design of the footwear’ to within the paragraph starting with ‘When it comes to foot measurements using the algometer…’.
Please revise this sentence: ”However, no study has specifically evaluated particular locations on the healthy foot.” Do you mean that no study has evaluated pressure pain thresholds of the feet in asymptomatic people?
This sentence should be stated earlier within the paragraph: ‘Moreover, pain measurement can be useful for tracking its progression over time or as a predictive tool for future outcomes’.
Please revise the aim as it isn’t clear: “Therefore, this study aimed to determine the agreement, intra-session and inter-session reliability, and validity of foot measurements using pressure pain threshold in a population of adults aged 55 to 75 years, not classified as healthy or unhealthy.” More specifically, I cannot see any work in the presented study to evaluate ‘validity’ Further, the wording ‘not classified as healthy or unhealthy’ is confusing. Do you mean asymptomatic adults?
Methodology: Is the study design suitable for answering the research question? Are the methods adequately described?
Within the section ‘Study Design and Participants’, please explicitly refer to reporting the study according to the GRRAS checklist (Kottner J, et al. J Clin Epidemiol. 2011;64(1):96-106), and cite the checklist and include a link to your supplementary file.
Within the GRAAS checklist supplementary file, there are multiple issues that need to be addressed. First, you appear to have misinterpreted the item ‘4. Specify the rater population of interest’. This item is referring to the assessor and instrument (e.g. podiatrist performing measurements). Also, it appears that you have misinterpreted item 7. ‘Describe the sampling method’ which refers to how you enrolled your participants (you did this via phone call to volunteers of a podiatry clinic), as well as item 9. ‘State whether measurements/ratings were conducted independently’ (as there was one rater/assessor, please detail in the manuscript how were they blinded to the value obtained each time they performed a measurement if the measurements were ‘independent’? otherwise, acknowledge this as a limitation within the discussion). These concepts are clearly described within the paper by Kottner et al. (2011). Other issues: The podiatrist was not blinded to measures, yet you say they were obtained ‘independently’ in your GRAAS checklist. Please revise this and be explicit in your manuscript that the assessor was not blind to their measures between sessions. Further, L334: you now state you used a 3,1 ICC model, but your GRAAS checklist reports a 1,1, model for sample size calculation (see item 6). Please address this inconsistency.

L281-283: I think it is critical to present the specific wording used to instruct participants when to advise they perceived pain. You refer to Fischer (1987) but I couldn’t see this study in your reference list. Please can you check this.
L291-296: please state your rationale for using these test sites on the feet.
L301-302: Can you be more specific in describing the podiatrist’s experience with use of the algometer. At present, you state ‘previous knowledge of the use’. Does this mean they never used it in clinical practice? Reporting the experience in more detail will help the reader understand the generalisability of the findings.
Results:
L348: Please delete ‘by an experienced podiatrists using a digital algometer, model Commander from JTECH’ as this is a repeat of the methods.

Discussion
L402, 408, 411: did the study evaluate ‘validity’? My thoughts are that it was solely reproducibility and you should not use the word validity. I think this is important to carefully check. From my perspective, to check for validity you would need to compare the PPT values to a gold standard measure (which you did not).
In referring to previous studies, the authors should focus on the findings related to reproducibility and how their study findings are similar or different, with possible reasons. At the moment, the authors have simply provided a study by study summary of previous work in this area, some results are irrelevant to the research question being studied (e.g. differences in PPTs between men and women is not relevant to the current study).
L461: I believe you are overstating the value of the tool (by using the word ‘outstanding’) particularly given the evidence and your study. Please temper your statement regarding its value.

Table 2:
Should the final line be the heel of the left foot? As it shows ‘R’? Unsure why there is reference to a ‘two sample t-test’?
Figure 2:
What do ‘lat’, ‘der’ and ‘izq’, ‘tipo’, ‘loc’, ‘C1’, ‘C5’ , ‘Pul’ ‘T’ mean? Please spell in ful are add to the description and explanation of abbreviations.
References:
Please carefully check all references have been included. For instance, I couldn’t see ‘Fischer (1987)

---

## Round 0.3 · Minor Revisions

We received mixed reviews for your revision. Before we can reach a decision I would like to ask you to consider the suggestions made by reviewer 2 regarding the GRRAS checklist, and to respond to the other comments

Reviewer 1 ·

Basic reporting

No comment

Experimental design

No comment

Validity of the findings

These types of psychophysical studies have merit in establishing normal ranges in both healthy participants and those with pathologies. The demographics provided, along with the narrow inclusion criteria, raise suspicion that this is not necessarily a healthy population, as BMI averages indicate that most were overweight. At these ages and weights, people tend to have diabetes and take many medications like blood thinners, beta blockers, etc. Also, at these ages, many patients can be cancer survivors after chemotherapy. With all these things not recorded and documented, it is difficult to ascertain the true nature of the population. They could be considered healthy, but they could also be considered heterogeneous and unhealthy. Please add this point as a limitation of the study.

Reviewer 2 ·

Basic reporting

Thank you for responding to my comments. The revised manuscript is much improved. Upon review of the revised mansucript, I have the following comments for your consideration.

Abstract:
Please remove this text ‘(ICC 0.76-0.91)’ from the final sentence as it is a repeat of the results in this section.

Materials and Methods
I don’t believe the authors responded to my suggestion in my previous review: ‘Within the section ‘Study Design and Participants’, please explicitly refer to reporting the study according to the GRRAS checklist (Kottner J, et al. J Clin Epidemiol. 2011;64(1):96-106), and cite the checklist and include a link to your supplementary file. I feel this is important to do as it sets a standard for subsequent researchers to follow best practice when performing reliability studies.
Of note in the attached supplementary file (GRAAS checklist):
Item 7 of checklist: the sampling method is the process of selecting a sample population from the target population. Please amend the checklist to correctly describe this (at the moment the procedures for obtaining the measurements are described here).

L148: You write: ‘This study aimed to validate a diagnostic test…’. I don’t think this is correct as it is not a diagnostic accuracy study. Please review.
L160 and L179: you state twice that participants provided informed consent. Please review this.
L192-3: please consider adding text regarding your justification as to why you chose the testing sites on the foot. Currently the justification is only within the author response document, but it would be good for the readers to be aware of the rationale.
L218: please amend word ‘statically’.
Discussion:
L302: replace the word ‘subjects’ with ‘participants’
L291-331 (paragraph 2 and 3): Thank you for revising the discussion. However, paragraph 2 and 3 lack focus. You have essentially summarised the results and implications from previous studies for most of it. Please review these paragraphs with a focus of stating the findings of your research and how they compare to previous relevant studies - those studies evaluating reliability of PPTs on the foot. Your focus was on reliability – so focus on previous studies that evaluated reliability of PPTs, and explain similarities and differences, and provide reasons for differences. This could occur more succinctly in one paragraph.
L333: Change the focus to ‘the existing evidence’ to ‘the findings of the current study support the …’. I think the focus should be on the implications for assessment and monitoring response to interventions, rather than ‘…design of new therapeutic strategies’. Please review this – as I am uncertain how the findings of a reliability study can inform the design of interventions.
Conclusions:
Please remove the text ‘(ICC: 0.76 to 0.91)’ from this section.

Table 2, 3 and 4:
Should ‘newton’ be ‘Newton’?

Experimental design

Please refer to section 1: Basic reporting

Validity of the findings

Please refer to section 1: Basic reporting

Additional comments

Please refer to section 1: Basic reporting

---

## Round 0.4 · accepted · Accept

You have adequately answered all comments, and the manuscript is now ready for the next steps of publication. Congratulations!